# Effects of Aging, Long-Term and Lifelong Exercise on the Urinary Metabolic Footprint of Rats

**DOI:** 10.3390/metabo10120481

**Published:** 2020-11-25

**Authors:** Anastasia Tzimou, Stefanos Nikolaidis, Olga Begou, Aikaterina Siopi, Olga Deda, Ioannis Taitzoglou, Georgios Theodoridis, Vassilis Mougios

**Affiliations:** 1Laboratory of Evaluation of Human Biological Performance, School of Physical Education and Sports Science, Aristotle University of Thessaloniki, 54124 Thessaloniki, Greece; stefanikol@hotmail.com (S.N.); renasiopi@gmail.com (A.S.); mougios@auth.gr (V.M.); 2Laboratory of Analytical Chemistry, Department of Chemistry, Aristotle University of Thessaloniki, 54124 Thessaloniki, Greece; olina_18@hotmail.com (O.B.); gtheodor@chem.auth.gr (G.T.); 3Biomic_Auth, Bioanalysis and Omics Lab, Centre for Interdisciplinary Research of Aristotle University of Thessaloniki, Innovation Area of Thessaloniki, 57001 Thessaloniki, Greece; oliadmy@gmail.com; 4Laboratory of Forensic Medicine and Toxicology, School of Medicine, Aristotle University of Thessaloniki, 54124 Thessaloniki, Greece; 5Laboratory of Physiology, School of Veterinary Medicine, Aristotle University of Thessaloniki, 54124 Thessaloniki, Greece; jotai@vet.auth.gr

**Keywords:** exercise, aging, urinary metabolites, liquid chromatography-mass spectrometry (LC/MS)

## Abstract

Life expectancy has risen in the past decades, resulting in an increase in the number of aged individuals. Exercise remains one of the most cost-effective treatments against disease and the physical consequences of aging. The purpose of this research was to investigate the effects of aging, long-term and lifelong exercise on the rat urinary metabolome. Thirty-six male Wistar rats were divided into four equal groups: exercise from 3 to 12 months of age (A), lifelong exercise from 3 to 21 months of age (B), no exercise (C), and exercise from 12 to 21 months of age (D). Exercise consisted in swimming for 20 min/day, 5 days/week. Urine samples collection was performed at 3, 12 and 21 months of life and their analysis was conducted by liquid chromatography-mass spectrometry. Multivariate analysis of the metabolite data did not show any discrimination between groups at any of the three aforementioned ages. However, multivariate analysis discriminated the three ages clearly when the groups were treated as one. Univariate analysis showed that training increased the levels of urinary amino acids and possibly protected against sarcopenia, as evidenced by the higher levels of creatine in the exercising groups. Aging was accompanied by decreased levels of urinary amino acids and signs of increased glycolysis. Concluding, both aging and, to a lesser degree, exercise affected the rat urinary metabolome, including metabolites related to energy metabolism, with exercise showing a potential to mitigate the consequences of aging.

## 1. Introduction

Over the past two centuries human life expectancy has more than doubled, resulting in a dramatic increase in the number of older individuals [1]. Aging comes with a multitude of physical and mental maladies, including common metabolic, inflammatory, cardiovascular and neurodegenerative diseases [2,3,4]. Therefore, the aforementioned increase in lifespan, without a concomitant increase in healthspan, will bring new challenges to health care systems, as, the years spent with poor health and disabilities at old age will increase. Therefore, studies on aging and healthy longevity are critical.

Physical activity and exercise are readily available to everyone and are simple, inexpensive, non-invasive and highly effective lifestyle interventions [5]. It is well documented that participation in sports and exercise programs leads to beneficial modifications in many interrelated biochemical pathways [6,7,8]. Regular exercise and physically active lifestyles are known to reduce the prevalence of metabolic syndrome and contribute to the prevention of chronic diseases such as diabetes, cardiovascular disease, obesity and cancer [9,10,11,12]. Exercise is also the only intervention to successfully prevent sarcopenia [13].

Metabolomics is a rapidly evolving field of life sciences that uses advanced analytical chemistry techniques in conjunction with sophisticated statistical methods to simultaneously measure and comprehensively characterize numerous metabolites [7,14,15]. The use of the power of metabolomics for the investigation of exercise- and aging-related biomarkers may aid in the early diagnosis and therapy of aging-related disorders [16], and our research team has been working in that direction over the past years [8,17,18,19,20,21,22,23]. Research into how long-term and, what is more, lifelong exercise impact healthy aging has so far produced limited information, one reason being the challenge of monitoring training humans through cohort studies for decades. This is why most data on the subject come from cross-sectional studies, which, however, do not permit distinction between cause and effect [24].

To address these difficulties, we have implemented lifelong exercise in laboratory rats, which have a much shorter life cycle than humans: they live about 2 years, become adult at 3 months and enter senility at about 21 months of age. Thus, we have examined the interaction of aging with long-term (during the 1st or 2nd half of life) and lifelong exercise training with regard to rat metabolism through metabolomic analyses. Results on the blood metabolome have already been presented [23]. Here, we present the analysis of the urinary metabolome through liquid chromatography-mass spectrometry (LC/MS). Our basic hypothesis was that the metabolic changes accompanying aging, especially in metabolites related to energy metabolism, would be modified by training.

## 2. Results

Fifty metabolites were identified by LC/MS in the urine samples, with four being in pairs (isoleucine-leucine and arabinose-xylose), since they could not be separated (Appendix A). The criteria for metabolite selection were detection in more than 80% of the samples, and noise-to-metabolite peak ratio of less than one-third.

### 2.1. Multivariate Analysis

Multivariate analysis through principal component analysis (PCA) and partial least-square discriminant analysis (PLS-DA) showed no discrimination between groups A (which exercised during the 1st half of their adult life, denoted as plain “1st half” from now on), B (the lifelong exercise group), C (non-exercising controls) and D (which exercised during the 2nd half of their adult life, referred to as “2nd half” from now on) at any of the three ages of urine sampling. In contrast, PCA (Figure 1) showed a clear discrimination between young, middle-aged and old rats. For comparative purposes, Figure 1 shows the PCA scores plots of metabolites normalized by either total ion current (TIC, Figure 1a) or creatinine (Figure 1b). Based on the similarities of the two normalization methods, all other datasets shown below were normalized by TIC. In order to get a more detailed view of the age differences, the three ages were examined in pairs (3 vs. 12, 3 vs. 21 and 12 vs. 21 months) with PCA and PLS-DA, and the results are presented in Figure 2. Scores plots showed a clear discrimination between ages again. According to the variable importance for the projection plot (VIP, Table 1), discrimination between 3 and 12 months was due to adenine, alanine, biotin, choline, mannitol, proline, pyruvate, tyrosine, and uridine, all of which decreased, as well as dimethylamine and tryptamine, both of which increased. Discrimination between 3 and 21 months was due to α-ketoglutarate, alanine, biotin, creatine, cytosine, guanine, isoleucine-leucine, mannitol, and proline, all of which decreased, as well as betaine and sarcosine, both of which increased. Lastly, discrimination between 12 and 21 months was due to adenine, choline, pyruvate, tyrosine, and uridine, all of which increased, as well as cytosine, dimethylamine, guanine, kynurenate, methylamine, trimethylamine-N-oxide (TMAO), and tryptamine, all of which decreased.

The illustration of the aforementioned findings through heat maps manifests the observed metabolite differences between ages and the correlation of specific metabolites that have affected the discrimination between groups. Specifically, samples are clearly discriminated between all three pairs of ages (Appendix A), based on Pearson’s correlation.

In order to find any additional information related to exercise, we examined samples collected at 12 and 21 months under the prism of whether the animals had exercised during the previous 9 months or not. To this end, groups A and B (which exercised during the 1st half) were compared with groups C and D (which did not exercise during the same period) at 12 months. The PLS-DA scores plot (Figure 3) showed a clear clustering between the exercising and non-exercising groups at 12 months. According to the VIP values, the metabolites responsible for this discrimination were creatine, cytidine, and cytosine, all of which increased, as well as thymidine, which decreased (Table 2). At the age of 21 months, no clustering was achieved between groups B and D (which exercised during the 2nd half) and groups A and C (which did not exercise during the same period). A summary of the model characteristics from the aforementioned PCA and PLS-DA multivariate analyses is presented in Table 3.

### 2.2. Univariate Analyses

The results of three-way ANOVA, comprising age (with repeated measures) × training or not during the 1st half × training or not during the 2nd half, are shown below. The subsequent descriptive statistics and the summary of the univariate analysis are presented in Appendix A.

#### 2.2.1. Amino Acids and Amino Acid Derivatives

An interaction of age and 1st half was seen in choline (*p* = 0.005, effect size 0.16). Particularly, the exercising groups showed a smaller decrease during the 1st half than the non-exercising groups.

An interaction of age and 2nd half was found in tryptophan (*p* = 0.006, effect size 0.15), betaine (*p* = 0.008, effect size 0.15) and alanine (*p* = 0.011, effect size 0.14). Specifically, all three metabolites increased in the groups that exercised during the 2nd half, whereas, in the groups that did not exercise, tryptophan did not change, while betaine and alanine decreased slightly.

An interaction of 1st and 2nd half was found in sarcosine (*p* = 0.007, effect size 0.26) and kynurenate (*p* = 0.005, effect size 0.27). During the 1st half, sarcosine decreased in the exercising groups more than in the non-exercising ones, whereas kynurenate increased in the non-exercising groups more than in the exercising ones. During the 2nd half, sarcosine increased, whereas kynurenate decreased in the exercising groups more than in the non-exercising ones.

A main effect of age was found in sarcosine (*p* < 0.001, effect size 0.52), alanine (*p* < 0.001, effect size 0.50), choline (*p* < 0.001, effect size 0.40), proline (*p* < 0.001, effect size 0.69), betaine (*p* < 0.001, effect size 0.38), isoleucine-leucine (*p* < 0.001, effect size 0.34), tyrosine (*p* < 0.001, effect size 0.25), tryptophan (*p* = 0.002, effect size 0.18), tryptamine (*p* < 0.001, effect size 0.41) and kynurenate (*p* < 0.001, effect size 0.37). Particularly, from 3 to 12 months, betaine, tryptamine and kynurenate increased, whereas proline, isoleucine-leucine, tyrosine and tryptophan decreased; from 12 to 21 months, sarcosine, alanine, choline and tyrosine increased, while tryptamine and kynurenate decreased; and, from 3 to 21 months, sarcosine, choline and betaine increased, whereas alanine, proline, isoleucine-leucine and tryptophan decreased.

#### 2.2.2. Carbohydrate and Lipid Metabolism

A main effect of age was seen in mannitol (*p* < 0.001, effect size 0.40), pyruvate (*p* < 0.001, effect size 0.62) and glucose (*p* < 0.001, effect size 0.30). Particularly, from 3 to 12 months, all three metabolites decreased; from 12 to 21 months, mannitol and pyruvate increased; and, from 3 to 21 months only mannitol showed a decrease.

#### 2.2.3. Krebs Cycle

An interaction of age and 2nd half was found in α-ketoglutarate (*p* = 0.005, effect size 0.16). Particularly, the exercising groups showed a decrease, whereas the non-exercising groups showed an increase. Also, a main effect of age was found in α-ketoglutarate (*p* < 0.001, effect size 0.32). Specifically, it showed a decrease from both 3 to 12 and from 3 to 21 months.

#### 2.2.4. Purine and Pyrimidine Metabolism

An interaction of age, 1st and 2nd half was noticed in guanine (*p* = 0.005, effect size 0.16) and cytidine (*p* = 0.005, effect size 0.21). Specifically, during the 1st half, guanine decreased in the exercising groups less than in the non-exercising ones, while cytidine remained the same in the exercising groups and increased in the non-exercising ones. During the 2nd half, guanine decreased less, whereas cytidine decreased more in the exercising groups than in the non-exercising ones.

An interaction of age and 1st half was seen in cytidine (*p* = 0.008, effect size 0.19). As mentioned above, the non-exercising groups showed an increase, whereas the exercising groups showed no change.

An interaction of age and 2nd half was seen in thymidine (*p* = 0.007, effect size 0.15). During the 2nd half, the exercising groups showed a decrease, whereas the non-exercising groups showed an increase.

A main effect of 1st half was found in adenine (*p* < 0.001, effect size 0.45) and cytidine (*p* < 0.001, effect size 0.44). Specifically, both metabolites had higher values at 12 months in the non-exercising groups than in the exercising groups.

A main effect of age was found in cytosine (*p* < 0.001, effect size 0.48), adenine (*p* < 0.001, effect size 0.47), guanine (*p* < 0.001, effect size 0.39), uridine (*p* < 0.001, effect size 0.59), thymidine (*p* = 0.002, effect size 0.15) and cytidine (*p* = 0.002, effect size 0.25). Particularly, from 3 to 12 months, cytosine and cytidine increased, whereas adenine, uridine and thymidine decreased; from 12 to 21 months, adenine and uridine increased, whereas cytosine, guanine and cytidine decreased; and, from 3 to 21 months, cytosine, adenine and guanine decreased.

#### 2.2.5. Gut Microbiome Metabolism

A main effect of 2nd half was found in TMAO (*p* = 0.007, effect size 0.26). Particularly, TMAO decreased in the exercising groups more than in the non-exercising ones.

A main effect of age was found in dimethylamine (*p* < 0.001, effect size 0.73) and TMAO (*p* < 0.001, effect size 0.45). Particularly, from 3 to 12 months dimethylamine increased, and from 12 to 21 months dimethylamine and TMAO decreased.

#### 2.2.6. Other Metabolites

An interaction of age and 1st half was seen in biotin (*p* < 0.001, effect size 0.27). Particularly, during the 1st half, it decreased in the exercising groups less than in the non-exercising ones.

An interaction of age and 2nd half was seen in creatine (*p* < 0.001, effect size 0.23). Particularly, the exercising groups showed an increase, whereas the non-exercising groups showed a decrease.

A main effect of age was found in creatine (*p* < 0.001, effect size 0.29), creatinine (*p* = 0.001, effect size 0.20), methylamine (*p* < 0.001, effect size 0.29) and biotin (*p* < 0.001, effect size 0.54). Specifically, from 3 to 12 months, creatinine increased, whereas creatine decreased; from 12 to 21 months, creatine increased, while creatinine and methylamine decreased; and, from 3 to 21 months creatine and biotin decreased.

## 3. Discussion

In the present study, we explored the effects of long-term and lifelong exercise of moderate intensity on the urinary metabolic footprint of rats through an LC/MS-based metabolomics method. Both aging and exercise altered more than half of the identified metabolites, involved in various metabolic pathways, including pathways of energy metabolism. The large effect sizes (all being above the benchmark of 0.138 and ranging from 0.14 to 0.73) showed that these effects were highly meaningful. However, the metabolic effects of aging seemed to be greater than those of exercise, in agreement with the findings of two previous studies, one on the urinary metabolome of female rats [22] and one on the blood metabolome of male rats [23]. Nevertheless, some larger effects of exercise seen in the former study [22] may be due to differences between sexes, such as in body weight, or due to the fact that male rats adapted more to training than female ones.

Recent reviews [9,14] have summarised many metabolomics-based studies that have investigated the acute and chronic effects of exercise on the metabolome of various biological fluids and tissues. However, studies on how lifelong exercise affects the metabolome is limited, possibly because surveilling living organisms from young through old age is a difficult task, especially when this involves studies lasting decades, as in humans. Hence, the main strength of the present research is that the interaction of lifelong exercise and aging was monitored on the same animals throughout most of their lifetime (18 out of about 24 months). The confounding effect of inter-individual variability is eliminated through the combination of a longitudinal design and a genetically homogenous population, contributing to the validity of the findings.

On the other hand, our study is limited by the relatively small sample size, which was mainly determined by the capacity of our facilities regarding housing and daily training of the animals. Hence, it is possible that a larger sample would make more differences to meet our fairly strict limit of statistical significance (5% FDR with α = 0.05). It should be pointed out, however, that the literature teems with metabolic studies using such small (or smaller) numbers of genetically uniform animals.

The results of both multivariate and univariate analyses are summarised in Figure 4, presenting the effects of training during both the 1st and 2nd half of life. The figure also helps to identify some interesting opposite effects of training and non-training on betaine, α-ketoglutarate, thymidine and creatine during the 2nd half. The effects of training and aging on urinary metabolites involved in major metabolic pathways, including pathways of energy metabolism, will be discussed below.

### 3.1. Amino Acids and Amino Acid Derivatives

Training seemed to increase amino acid concentrations, particularly during the 2nd half, while aging during the 1st half and from 3 to 21 months seemed to decrease amino acid concentrations. Specifically, aging resulted in an expected decrease in two glucogenic amino acids, alanine and proline, from young age to midlife, while training resulted in an increase from midlife to elderhood. Both alanine and proline are amino acids that have been negatively associated with aging [4,25]. 

Sarcosine increased with aging, and exercise resulted in a decrease during the 1st half, although during the 2nd half the exercising groups exhibited a greater increase. Betaine increased from young age to midlife. From midlife to elderhood, training resulted in an increase, whereas no training resulted in a decrease. Sarcosine and betaine are intermediates in the synthesis of glycine from choline. Circulating betaine has been positively associated with physical activity [26] and an increase has been reported after weight reduction [27]. Choline decreased during the 1st half, with the exercising groups exhibiting a smaller decrease, and increased during the 2nd half. However, choline excretion has been found increased after 18 months of exercise training in older men [28].

Aging resulted in decreases in the urinary concentrations of the paired branched-chain amino acids (BCAA), isoleucine and leucine. Although the effects of aging on BCAA metabolism are not clear, there are studies that have found decreased BCAAs in the blood of elderly subjects and have attributed this to reduced muscle activity [29]. Two aromatic amino acids, tryptophan and tyrosine, decreased from young age to midlife, whereas an aromatic amino acid derivative, kynurenate, increased during the same period. These findings suggest increased aromatic amino acid catabolism. Moreover, tryptophan decreased more in the exercising groups than in the non-exercising ones. From midlife to elderhood, tyrosine showed an increase, whereas kynurenate and tryptamine showed a decrease. The decrease of kynurenate in the exercising groups was bigger than that in the non-exercising ones. Tryptophan did not show any changes due to aging during the same period, although an exercise-induced increase was noticed. Plasma concentrations of tryptophan and tyrosine are negatively associated with aging [4,25,30,31]. The concomitant increase in kynurenine, a precursor of kynurenate, has been associated with inflammatory diseases [32,33]. In a study with older men, urinary tryptophan increased following 18 months of training [28]. This increase was attributed to increased serotonin production, for which tryptophan is a precursor, and was associated with exercise training. Following training, tryptophan levels increased, albeit in blood of aged mice [33], which was also the case in the present study. Increases in blood tryptophan and tryptophan metabolites have also been observed following both acute exercise and training and have been positively associated with maximal oxygen uptake, VO_2_ max [30,34,35]. 

### 3.2. Carbohydrate and Lipid Metabolism

The observed decrease in urinary glucose and pyruvate during the 1st half of life implies decreased flow of carbohydrates through glycolysis. By contrast, an important finding related to energy metabolism was the increase in pyruvate during the 2nd half, which implies accelerated glycolysis. Other studies have also found an increase in glycolytic rate with aging [36,37,38]. Mannitol decreased from young age to midlife, whereas it increased from midlife to elderhood. However, the levels at 3 months were higher than those at 21 months, suggesting an aging-related decrease. These fluctuations are possibly the results of either differences in food intake or impairment in intestinal permeability. This is, once more, supported by Valentini et al. [39], who also found decreased urinary mannitol in aged adults. 

Another finding related to energy metabolism was the decrease in acetylcarnitine from midlife to elderhood according to both multivariate and univariate analyses. A decrease in urinary acetylcarnitine concentration was also noticed in older individuals [40] and aged female rats [22]. Since acetylcarnitine is formed when there is abundance of acetyl CoA (due to increased carbohydrate and lipid oxidation [41]), its decrease may suggest a decrease in these two metabolic pathways with aging.

α-Ketoglutarate decreased from young age to midlife and from young age to elderhood, suggesting an age-related decrease. Decreased concentrations of several Krebs cycle metabolites in urine have been associated with aging and diminished mitochondrial function [42]. 

### 3.3. Purine and Pyrimidine Metabolism

Adenine decreased from young age to midlife and from young age to elderhood, although there was an increase from midlife to elderhood. No studies were found referring to the effects of aging and/or exercise on urinary purine concentrations. However, a study on plasma purine concentrations found that they increased with age, representing the depletion of the skeletal muscle adenine nucleotide pool [43]. Guanine, another purine, was found decreased in the non-exercising groups during both the 1st and 2nd half. 

Cytosine and cytidine increased from young age to midlife, while thymidine and uridine decreased. From midlife to elderhood cytosine and cytidine decreased, while thymidine and uridine increased. Pyrimidine metabolites are essential for the synthesis of DNA, RNA, lipids and carbohydrates, and their downregulation has been linked to aging and diseases, such as Alzheimer’s disease and immunodeficiency [44]. Cytosine also exhibited an exercise-related increase during the 1st half. Increases in pyrimidine nucleotides have been observed after both acute and prolonged exercise regimes [45]. Exercise also resulted in a decrease in thymidine during the 1st half, and only the non-exercising groups exhibited an increase in cytidine during the same period. During the 2nd half, the exercising groups exhibited a decrease in cytidine and thymidine, whereas the non-exercising groups showed an increase in thymidine.

### 3.4. Gut Microbiome Metabolism

From young age to midlife, dimethylamine and TMAO (two metabolites associated with gut microbiome metabolism) increased, whereas from midlife to elderhood they decreased. Exercise during the 2nd half accentuated the decrease in TMAO. TMAO decreased following exercise in other studies, and this decrease has been related to health benefits [8,46]. Aging has been reported to cause higher concentrations of dimethylamine and TMAO, and this increase is speculated to result from impairments in renal function [47,48] and/or muscle composition [30]. Dimethylamine was also found elevated in patients with hypertension and cardiac dysfunction [28], and TMAO has been associated with reduced survival rates in conditions such as heart failure [14]. TMAO is also a product of choline metabolism; thus, the decrease in choline and the concomitant increase in TMAO during the 1st half may support the hypothesis of increased choline catabolism with aging, as mentioned above. 

### 3.5. Other Metabolites

Creatine decreased from young age to midlife, whereas creatinine increased during the same period. This implies increased conversion of creatine to creatinine. During the 2nd half, however, no changes were seen in creatine, although creatinine showed a decrease. The excretion of creatinine has been reported to reflect muscle mass [49]. Likewise, in another study [50], urinary creatinine increased in rats between the 4th and 18th months of age and then decreased up until the 24th month. The findings of the present study agree with the aforementioned studies. Moreover, during the 2nd half, creatine increased in the exercising groups and decreased in the non-exercising ones. This finding may show a beneficial effect of exercise in protecting against the age-related loss of muscle mass, known as sarcopenia. Aging and sarcopenia have been linked with a decline in muscle creatine, which is exacerbated with physical inactivity [51].

Methylamine excretion decreased from midlife to elderhood in the present study. Urinary methylamine also decreased between the 4th and 24th months [50] and between the 5th and 21st months of age in female rats [22]. Thiamine and biotin, two B vitamins, decreased with aging. Specifically, thiamine decreased from young age to midlife, whereas biotin decreased from young age to elderhood. There are indeed alterations in nutrient availability due to aging, including altered vitamin uptake [52]. Neither thiamine nor biotin excretion was found altered with age in an older study, with biotin exhibiting an increase at a younger age and then remaining stable until elderhood [53]. In another study, thiamine was found deficient among older adults and was associated with heart failure [54]. During the 1st half, biotin decreased in the non-exercising groups. No studies could be found to compare this finding with; however, it may suggest that physical inactivity exacerbates the age-related impairment in vitamin availability.

## 4. Materials and Methods

### 4.1. Animals, Study Design, and Ethics

Thirty-six male Wistar rats were obtained at the age of 3 months (estimated onset of adulthood) from the Lab Animal facility of the Laboratory of Physiology, School of Veterinary Medicine, Aristotle University, Thessaloniki. We decided on male rats to prevent any misinterpretation of our results due to factors associated with the female reproductive cycle and menopause. The animals were caged in groups of 3 at 22 °C with relative humidity at 50%. Rodent chow and water were accessed freely by the animals, and their weight and food intake were checked on a monthly and weekly basis, respectively, on a digital scale to the nearest gram. The results on body weight and food intake are presented in our recently published work [23].

Four equal groups (*n* = 9 each, Figure 5) were created by dividing the rats in a random manner: groups A exercised from the 3rd to 12th month of life (1st half); group B exercised from the 3rd to 21st month of life (lifelong exercise); group C did not exercise; and group D exercised from the 12th to 21st month of life (2nd half). Seven animals died, apparently from natural causes, during the 2nd half of the study, and the 29 animals left at 21 months were distributed as follows: Group A, *n* = 7; group B, *n* = 9, group C, *n* = 8; and group D, *n* = 5. All procedures were conducted in accordance with the Directive 2010/63/EU and the Presidential Decree No. 56/2013 of Greek legislation for care and use of laboratory animals and were approved by the Department of Rural Economy and Veterinary Medicine, Prefecture of Central Macedonia, Hellenic Republic (Code no. EL54BIO10, Protocol no. 449161/4835).

### 4.2. Exercise Training

The rats exercised by swimming individually for 20 min/day, 5 days/week. This type of training was chosen, since it is free from injuries, pleasant to rats, and does not require the use of electrical shocks, as is the case with treadmill running, to make them exercise. Details on the lifelong training and acclimatization protocol are provided in our recently published work [23]. The effect of ambient temperature on metabolism was controlled for by keeping the non-exercising rats in shallow water of the same temperature and for the same time as the exercising ones.

### 4.3. Sampling

Urine samples were collected at the ages of 3 (36 animals), 12 (36 animals) and 21 months (29 animals) which corresponded to the beginning, middle and end of the study, respectively, generating a total of 101 samples. The day before the acclimatization day the pre-training samples were collected, while post-training samples were collected three days after the last training session to avoid any acute effects on the urinary metabolome. Samples were collected at around the same time of the day (with a deviation of up to one hour) for each animal. To collect urine samples, a slight pressure was applied to the pubic area, and the rat was left to urinate on a clean surface. Urine was collected with the use of a pipette and was immediately placed in a tube and stored at −80 °C until analysis. It has been found that urinary samples remain stable for up to 6 months at this temperature [55], with no research on longer storage periods, to the best of our knowledge. There is research, however, on blood samples that shows stability for up to 30 months [56].

### 4.4. LC/MS Analysis

The LC/MS analysis of urine metabolites was based on the method described by Virgiliou et al. [57]. Briefly, 20 μL of urine were placed in a tube, and 60 μL of ice-cold acetonitrile were added. A quality control (QC) sample, consisting of 10 μL of all samples, was also prepared to check the method’s precision and accuracy. 20 μL of this mixture were removed into a separate tube, and 60 μL of acetonitrile were added. All tubes were then vortexed for 5 min and centrifuged at 18,000× *g* for 15 min at 6 °C. The supernatants were removed and filtered through 13-mm syringe filters before being transferred to LC/MS inserts. They were then sealed and loaded in the autosampler tray that was maintained at 10 °C for the duration of the analysis. The QC sample was run in triplicate at the beginning of each sample sequence and then every ten experimental samples. Apart from that, a mix of standards, which included the metabolites of the applied method, at three different concentrations was analysed before and after analysis of the samples to test the system’s analytical stability. Liquid chromatography and mass spectrometry were performed on an ACQUITY liquid chromatograph and a Xevo TQD MS System (Waters, Milford, MA, USA), equipped with the Mass-Lynx software, using an ACQUITY HILIC, BEH amide column (2.1 mm × 150 × mm, 1.7 μm). The method and equipment are described in detail by Virgiliou et al. [57]. For the normalisation of the data, TIC was used, in which the signal of each metabolite in a sample was divided by the sum of signals of the ions detected in the sample; for practical pursposes, this was then multiplied by 1000. 

This normalisation method was chosen, since, according to Veselkov et al. [58], it performs well in urinary samples and corrects for urine volume differences when relative concentration is measured. Moreover, our group has previously used both total area (identical to TIC) and median fold change, which are the two mostly used normalisation methods, in rat serum and found that the former method performed better than the latter [19]. Furthermore, we tested normalisation by creatinine in the present study and found that the pattern of the graphs produced by the mutlivariate statistics and the majority of the univariate analyses results were similar with those produced after normalisation by TIC. A final reason for choosing normalisation by TIC was that urinary creatinine has been shown to vary with age [40,50,59] and exercise [46,59], the two variables investigated in the present study.

### 4.5. Statistical Analyses 

Principal component analysis (PCA) and partial least-square discriminant analysis (PLS-DA) by use of the SIMCA-P software (version 11.5, Umetrics, Malmö, Sweden) were performed for the multivariate analysis of the metabolite data. The scaling method applied was UV, where standard deviation is the scaling factor. The scores plots produced from the aforementioned analyses showed whether there was a discrimination between groups. To examine the correlation of specific variables with the discrimination between groups a VIP plot and list were created. The ‘important’ variables were those with values equal to or larger than one, generated by the VIP plot and list, whereas ‘unimportant’ variables were those with values lower than one. Lastly, a validation check with 100 permutations was performed for every PLS-DA model, which compared the goodness of fit of the original model with the quality of fit of several models (depending on the number of permutations set). Outliers appearing in multivariate analyses were not treated so as to keep data processing unbiased. Nevertheless, they did not seem to affect the scores plots produced. 

Complementary to the multivariate analysis described above, univariate analysis was conducted by three-way analysis of variance (ANOVA), that is, age (with repeated measures) × training or not during the 1st half of life × training or not during the 2nd half of life in SPSS (version 25, IBM Statistics, Armonk, NY, USA). Data were checked for normality of distribution, and only those that passed the Shapiro-Wilk test were analysed. Sphericity was also checked with Mauchly’s test and when it was not met the correction of Huynh-Feldt was used. Type I error was controlled for by correcting significant main effects and interactions with a 5% adjusted false detection rate (FDR) according to Benjamini and Hochberg [60]. Important metabolites were also assessed using receiver operating characteristic (ROC) curve analysis. The area under the ROC curve (AUC-ROC) is an indicator of the ability of important metabolites to discriminate groups. Values over 0.7 suggest an acceptable discrimination. The representations of the changing patterns of metabolic profiles by heat maps, AUC-ROC, and log_2_ fold change were created and calculated with MetaboAnalyst v. 4.0 (www.metaboanalyst.ca; Chong et al. [61]). The level of significance was set at α = 0.05. Effect sizes for main effects and interactions were estimated by calculating partial eta squared and were classified as small (0.01 to 0.058), medium (0.059 to 0.137) or large (0.138 or higher), according to Cohen [62]. 

## 5. Conclusions

As a whole, the urinary metabolic footprint of rats changed with age but not with long-term or lifelong exercise, although many individual changes of metabolites were seen in metabolic pathways related to exercise metabolism. Notably, training resulted in increases in urinary amino acids, especially during the 2nd half, whereas aging had the opposite effect. Aging was also associated with signs of increased glycolysis from midlife to elderhood. We found evidence for aging-related impairment in intestinal activity and permeability, as well as vitamin availability, with physical inactivity exacerbating these negative effects. A potentially important finding was the opposite changes of creatine during the 2nd half in the exercising and non-exercising groups (increase and decrease, respectively), suggestive of a protective effect of exercise against sarcopenia. Our findings support the potential of exercise to mitigate the consequences of aging.

## Figures and Tables

**Figure 1 metabolites-10-00481-f001:**
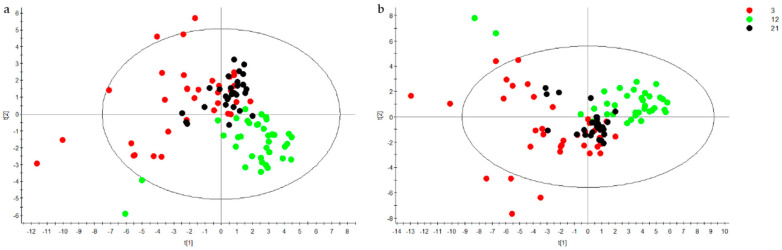
PCA scores plots of urine samples from young (3-month old, red dots), middle-aged (12-month old, green dots) and old rats (21-month old, black dots), normalized by total ion current (**a**) and creatinine (**b**). Ellipse represents 95% confidence intervals of Hotelling’s T^2^ distribution.

**Figure 2 metabolites-10-00481-f002:**
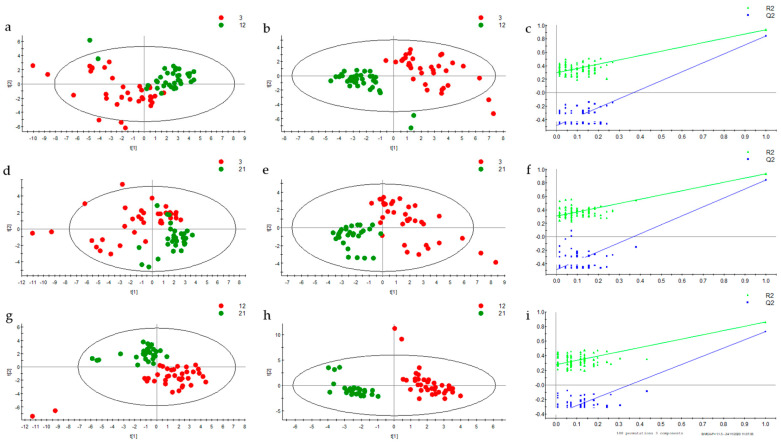
(**a**) PCA scores plot between 3 (red) and 12 months (green), (**b**) corresponding PLS-DA scores plot, (**c**) corresponding permutations plot, (**d**) PCA scores plot between 3 (red) and 21 months (green) (**e**) corresponding PLS-DA scores plot, (**f**) corresponding permutations plot, (**g**) PCA scores plot between 12 (red) and 21 months (green), (**h**) corresponding PLS-DA scores plot, (**i**) corresponding permutations plot.

**Figure 3 metabolites-10-00481-f003:**
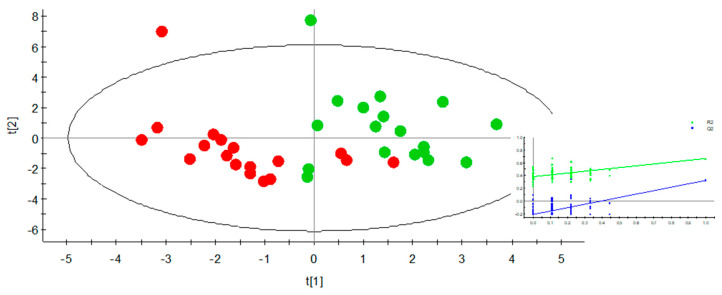
PLS-DA scores plot of urine samples at 12 months of age between the groups that had exercised (A and B, green) and the groups that had not exercised during the 1st half (C and D, red) with permutations plot as insert.

**Figure 4 metabolites-10-00481-f004:**
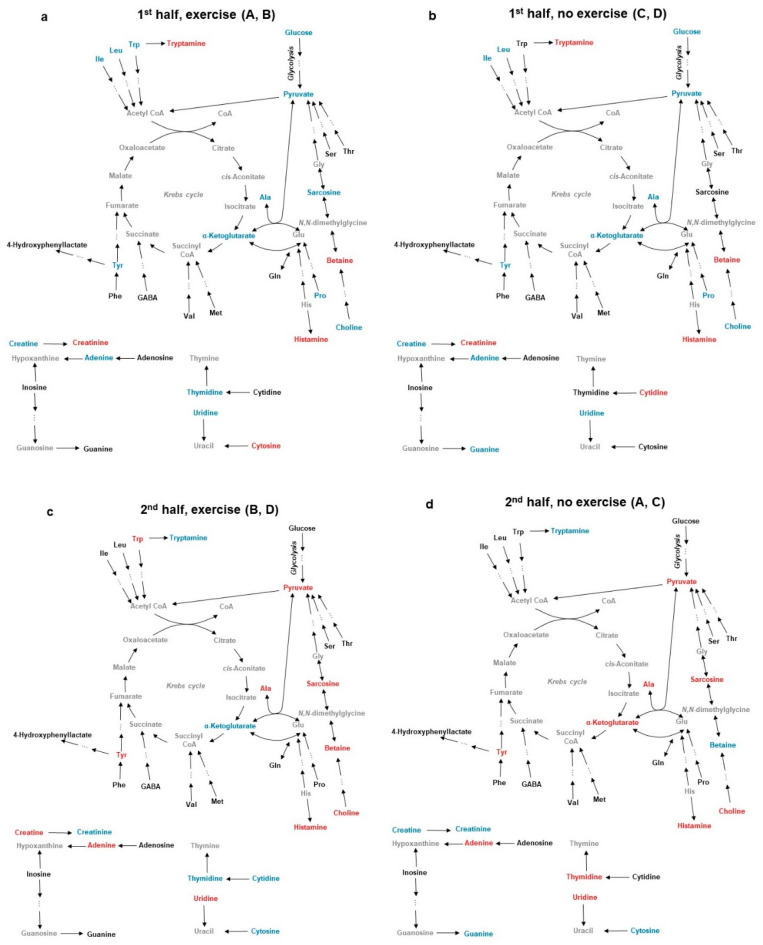
Map of metabolic pathways with highlighted urinary metabolites that changed with age and/or exercise training in (**a**) the exercising groups during the 1st half, (**b**) the non-exercising groups during the 1st half, (**c**) the exercising groups during the 2nd half, and (**d**) the non-exercising groups during the 2nd half of life. Metabolites in red, blue, or black increased, decreased, or did not change, respectively. Metabolites in grey were not identified.

**Figure 5 metabolites-10-00481-f005:**
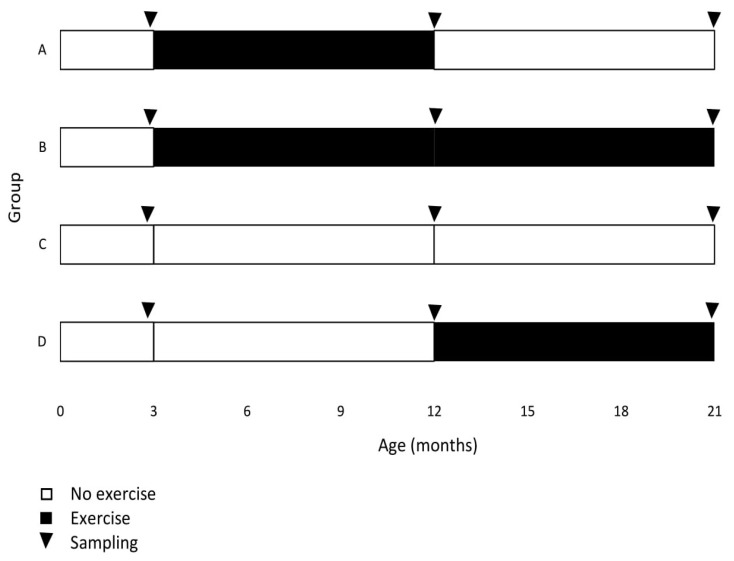
Study design showing the interventions on the four groups of rats. Solid bars denote exercise training, open bars denote no exercise, and arrowheads denote urine sampling.

**Table 1 metabolites-10-00481-t001:** Metabolites contributing to the discrimination between age groups. (Note: VIP: variable importance for the projection plot; AUC-ROC: area under the curve-receiver operating characteristic; log_2_FC: log_2_ fold change. All results presented had *p* values < 0.001.)

	3 vs. 12 Months			3 vs. 21 Months			12 vs. 21 Months		
Metabolites	VIP	AUC-ROC	log_2_FC	VIP	AUC-ROC	log_2_FC	VIP	AUC-ROC	log_2_FC
Adenine	1.75	0.94	−1.34				1.77	0.90	1.09
Alanine	1.25	0.91	−1.94	1.24	0.77	−0.63			
Betaine				1.35	0.81	0.29			
Biotin	1.21	0.82	−1.57	2.00	0.98	−2.00			
Choline	1.37	0.87	−2.05				1.64	0.87	0.32
Creatine				1.33	0.81	−0.54			
Cytosine				1.60	0.85	−0.65	1.68	0.89	−0.86
Dimethylamine	1.34	0.85	0.19				1.53	0.87	−0.19
Guanine				1.49	0.82	−0.77	1.08	0.73	−0.48
Isoleucine-leucine				1.39	0.88	−1.19			
Kynurenate							1.06	0.78	−0.81
Mannitol	1.64	0.91	−1.19	1.10	0.76	−0.56			
Methylamine							1.01	0.76	−0.65
Proline	1.68	0.94	−0.96	1.86	0.92	−0.76			
Pyruvate	1.65	0.90	−0.89				1.71	0.87	0.83
Sarcosine				1.32	0.82	0.84			
TMAO							1.19	0.74	−0.26
Tryptamine	1.12	0.78	1.39				1.10	0.73	−0.62
Tyrosine	1.07	0.81	−1.10				1.21	0.84	1.16
Uridine	1.71	0.94	−1.01				1.86	0.93	0.97
α-Ketoglutarate				1.50	0.82	−0.49			

**Table 2 metabolites-10-00481-t002:** Metabolites contributing to the discrimination between the exercising and non-exercising groups at 12 months. (See Table 1 for explanations.)

Metabolites	VIP	*p*-Value	AUC-ROC	log_2_FC
Creatine	1.04	0.026	0.67	0.04
Cytidine	1.74	<0.001	0.77	1.11
Cytosine	1.11	0.043	0.61	0.17
Thymidine	1.86	0.012	0.79	−0.96

**Table 3 metabolites-10-00481-t003:** Summary of model characteristics from PCA and PLS-DA multivariate analyses.

Comparisons	Analysis	R^2^X	R^2^Y	Q^2^Y
3 vs. 12 vs. 21 months	PCA	0.535		0.199
3 vs. 12 months	PCA	0.689		0.254
	PLS-DA	0.531	0.941	0.847
3 vs. 21 months	PCA	0.381		0.171
	PLS-DA	0.43	0.862	0.733
12 vs. 21 months	PCA	0.381		0.18
	PLS-DA	0.446	0.956	0.914
AB vs. CD (12 months)	PLS-DA	0.363	0.671	0.325

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
