# Peer review of "Effects of Aging, Long-Term and Lifelong Exercise on the Urinary Metabolic Footprint of Rats"

_metabolites, 2020, doi:10.3390/metabo10120481_

Round 1

Reviewer 1 Report

Tzimou et al investigated the effect of aging and physical exercise on the urine metabolome. The study employed male Wistar rats of various ages with or without a physical exercise regime that consisted of swimming 5 days/week, 20 minutes/day.

Data on blood metabolome for this study was previously published by the authors elsewhere (Tzimou, A.; Benaki, D.; Nikolaidis, S.; Mikros, E.; Taitzoglou, I.; Mougios, V. Effects of lifelong exercise and aging on the blood metabolic fingerprint of rats. Biogerontology 2020 (online ahead of print)).

Stratification by age or type of physical exercise did not show discrimination between conditions, however, when all study subgroups were unified as one, a discriminatory effect by age was observed. Further, the authors identified elevation of urinary amino acids and creatine with physical exercise. Older animals exhibited decreased urinary amino acids and fingerprints of elevated glycolysis. The authors conclude that both aging and physical exercise impact energy metabolism.

The study is of interest to the audience of Metabolites. The study protocol shows a robust experimental design and is described in detail. The statistical analysis is adequate and explained with sufficient detail.

There are a few points that require attention prior to consideration for publication:

  • Normalization of urine metabolites, page 13, lines 392-394: The authors used total ion count to normalize metabolite peak areas. This is a widely used generic approach that works well for biofluids such as blood or cerebrospinal fluid, but is less appropriate for urinary metabolites due to massive effects caused by hydration state and possible variations in renal function. The latter being of particular relevance in a study of aging. Urinary metabolites are by default normalized by the concentration (or peak area if performing semi-quantitative metabolomics like in this study) of biomarker of renal excretion creatinine. For the metabolites that were examined individually, for example, excretion of BCAA, TMAO, among others, do the statistical differences (3-way ANOVA) remain if peak areas are now normalized by the respective peak areas of creatinine?

  • Interpretation of creatinine levels, section 3.5., lines 317-322: This section is not accurate. The concentration of creatinine in urine varies mainly with hydration status, and secondarily with renal function. The contribution of muscle waste to urinary creatinine is at most, not a major contributor to the total concentration of creatinine in urine. The assumption of increased creatinine as a result of aging or muscular activity cannot be assessed by means of the metabolomic studies presented herein. In particular, it is very hard to accurately control water intake by animals. Water and food are typically ad libitum. One way to examine creatine and creatinine is by isotopic labeling and assessment of renal clearance. I suggest modifying or deleting this section completely, and instead present urinary metabolites normalized by creatinine abundance such to be able to draw physiological meaning.

  • Univariate analysis: the authors described the main findings well in the main text but summary tables would be best to visualize the individual variances as well as the interactions. If there is no space in the main text for additional tables, please, include them in Supplemental Data. There is a mention to such tables in the main text (page 5, lines 125-127) but I could not find them in the Supplemental Data file that is available for download.

  • Summary of main individual metabolite changes: the authors profiled an interesting set of metabolites and discussed these findings in the results and discussion sections. It would be great if they could also create graphs highlighting selected metabolites normalized by creatinine. For example, the authors could select metabolite subsets that appear to be the ‘main drivers’ in the discrimination by age, or by physical exercise, etc. and focus the discussion on these particular metabolite subsets. 

Reviewer 2 Report

The authors present a manuscript studying the metabolome of urinary samples from rats in which the effect of life-long exercise is studied.

The manuscript seems to be a follow-up of similar research (conducted on blood samples) submitted to the Journal of Biogerontology.

I have several major issues with the research presented:

1) how sure are the authors they are not identifying differences in the urinary metabolome based on the storage of the samples, presuming the different batches (age specific cohorts) were not harvested at the same time? For this QC's of the cohorts (based on age) with appropriate internal standards should have been prepped and run at different times during the experiment to represent the freeze conditions of the samples.

2) normalisation by TIC for the urine can be improved, for instance the work presented in "Optimized Preprocessing of Ultra-Performance Liquid Chromatography/Mass Spectrometry Urinary Metabolic Profiles for Improved Information Recovery. Analytical Chemistry, 2011, 83(15): 5864–5872."

3) how can the authors correct for urine volume, this is based on the sample collection which consisted of applying pressure to the pubic area of the animals.

4) were there no other physical readouts for the groups such as BMI, fat content etc?

5) it is stated that 50 metabolites were identified, was the PCA based on this set of 50 metabolites? and of these 50 metabolites, 21 showed a significant change? This seems to me that other factors than biology play a role here (unless the author have been extremely lucky). A solution is to do untargeted approaches and using all features generated by the MS (even if you can only identify 50 of them), I believe this would make the PCA plots more informative.

6) the description of the targets from table 1 in section 2.2 does not make sense to me, this is a very descriptive attempt to correlate targets to the biological sets. For me, there is no added value here and the entire section could be removed.

2.12.0.0

Round 2

Reviewer 1 Report

The revised version of the manuscript incorporates suggested changes in an adequate manner. Concerning specific points that were discussed in this review, please, adjust the following points prior to consideration for publication: 

1) While it is correct that creatinine may vary with age and exercise, this is not always the case and therefore, not a reason to dismiss its use as a metabolite for normalisation purposes. Indeed, creatinine concentration in urine is extremely stable and almost exclusively affected by renal disease and hydration status. As such, creatinine is the gold-standard used in medicine to normalize all urinary metabolites currently used in clinical chemistry, regardless of the age of the patient or how much exercise is involved in his/her lifestyle. The results of PCA analysis after normalisation by creatinine as provided by the authors in this revision are reassuring of the robustness of the findings in this study. Therefore, this PCA graph should be shown in the manuscript. One possibility is to add it to Figure 1 (as new panel B) for comparative purposes (Panel A: PCA of metabolites normalised by TIC, and panel B: PCA of metabolites normalised by creatinine) to clearly show the basis for the decision of normalisation by TIC. The authors can state that based on similarities of the two normalisation methods, all other datasets shown below were normalised by TIC (hence, all other figures in the manuscript remain unchanged). 

2) Materials and Methods, section  4.4. LC-MS/MS analysis: references in new paragraph covering normalisation were provided in the response to the reviewer but are missing in the manuscript. Please, insert new references in areas currently shown as [...]. 

Also, wording of the last sentence: "A final reason for choosing normalisation by TIC was that urinary creatinine has been shown to change with age [...] and exercise [...]." 

Please, add the following words to complete the statement: 

"A final reason for choosing normalisation by TIC was that urinary creatinine has been shown to vary with age [...] and exercise [...], the two variables investigated in the present study."

Author Response

1) While it is correct that creatinine may vary with age and exercise, this is not always the case and therefore, not a reason to dismiss its use as a metabolite for normalisation purposes. Indeed, creatinine concentration in urine is extremely stable and almost exclusively affected by renal disease and hydration status. As such, creatinine is the gold-standard used in medicine to normalize all urinary metabolites currently used in clinical chemistry, regardless of the age of the patient or how much exercise is involved in his/her lifestyle. The results of PCA analysis after normalisation by creatinine as provided by the authors in this revision are reassuring of the robustness of the findings in this study. Therefore, this PCA graph should be shown in the manuscript. One possibility is to add it to Figure 1 (as new panel B) for comparative purposes (Panel A: PCA of metabolites normalised by TIC, and panel B: PCA of metabolites normalised by creatinine) to clearly show the basis for the decision of normalisation by TIC. The authors can state that based on similarities of the two normalisation methods, all other datasets shown below were normalised by TIC (hence, all other figures in the manuscript remain unchanged). 

Response: The figure has been added as fig. 1b, with a description in lines 81-84.

2) Materials and Methods, section  4.4. LC-MS/MS analysis: references in new paragraph covering normalisation were provided in the response to the reviewer but are missing in the manuscript. Please, insert new references in areas currently shown as [...]. 

Response:  We apologise for the omission.  All references have been added as # 55, 56, 58 and 59.

Also, wording of the last sentence: "A final reason for choosing normalisation by TIC was that urinary creatinine has been shown to change with age [...] and exercise [...]." 

Please, add the following words to complete the statement: 

"A final reason for choosing normalisation by TIC was that urinary creatinine has been shown to vary with age [...] and exercise [...], the two variables investigated in the present study."

Response: The suggested change has been made, lines 413-415.

Thank you once more for your constructive comments and for contributing to the improvement of our manuscript.

Reviewer 2 Report

I believe the authors have done their best to adjust the text where needed and replied to my concerns adequately.

Author Response

I believe the authors have done their best to adjust the text where needed and replied to my concerns adequately.

Response: Thank you once more for your constructive comments and for contributing to the improvement of our manuscript.